# A New Approach for the Development and Optimization of Gluten-Free Noodles Using Flours from Byproducts of Cold-Pressed Okra and Pumpkin Seeds

**DOI:** 10.3390/foods12102018

**Published:** 2023-05-16

**Authors:** Ebru Aydin, Sebahattin Serhat Turgut, Sedef Aydin, Serife Cevik, Ayse Ozcelik, Mehmet Aksu, Muhammed Mustafa Ozcelik, Gulcan Ozkan

**Affiliations:** 1Department of Food Engineering, Faculty of Engineering, Suleyman Demirel University, 32200 Isparta, Turkey; ebruaydin@sdu.edu.tr (E.A.); serhatturgut@sdu.edu.tr (S.S.T.); sedefaydin03@gmail.com (S.A.); aysebiyikli@sdu.edu.tr (A.O.); gmaksu@gmail.com (M.A.); ozcelik.m.mustafa@gmail.com (M.M.O.); 2Department of Food Processing, Gelendost Vocational High School, Isparta University of Applied Sciences, 32900 Isparta, Turkey; serifecevik@isparta.edu.tr

**Keywords:** valorization, pumpkin, okra, seed, noodle, optimization with R

## Abstract

The significant protein and dietary fiber content of cold-pressed pumpkin (PSF) and okra (OSF) seed byproducts are well-known. However, their impact on noodles’ nutritional quality has never been studied. For the first time, noodle formulation was developed employing a genetic algorithm in the R programming language to achieve the most optimal sensory attributes as well as nutritional composition, color, cooking, and textural properties. The optimized noodle formulation was detected for OSF, PSF, gluten-free flour, salt, and egg with the following amounts: 11.5 g, 87.0 g, 0.9 g, 0.6 g, and 40 g, respectively, with 10.5 mL of water. The total protein (TP%), total fat (TF%), total carbohydrate (TC%), total dietary fiber content (TDF%), ash (%), total phenolic content (TPC mg GAE/100 g), and ABTS (%) of PSF were found to be 39%, 17%, 7%, 18%, 3%, 19%, and 48%, respectively, whereas for OSF, 33%, 8%, 21%, 32%, 5%, 16%, and 38%, respectively, were detected. In addition, TP (42.88%), TF (15.6%), ash (5.68%), TDF (40.48%), TPC (25.5 mg GAE/100 g), and ABTS (70%) values were obtained for the noodles. Consequently, the valorization of the cold oil press industry’s byproducts may be used as ingredients that add high value to gluten-free protein and fiber-rich noodle production, and they may gain interest from both processors and consumers.

## 1. Introduction

Food byproduct valorization will benefit both humanity and the environment by reducing waste in landfills and lowering the cost of raw materials. Seed flour is a residue of edible seed oil extraction (ESOE) that is commonly discarded as waste (cake/flour). Although consumer interest in natural and healthy functional foods is increasing, authorities are paying growing attention to environmental concerns and food sustainability. Recent studies used ESOE byproducts as functional ingredients for the enrichment of foods such as cake, salad dressing, and bread [1,2]. Nowadays, the valorization of these ESOE byproducts is getting attention due to their rich protein, fiber, antioxidant, mineral, and fatty acid content. Therefore, they are accepted as a great source for developing functional foods and also to provide a reduction in byproduct release [1,2,3,4]. In this study, it was aimed to optimize a formulation of gluten-free noodles by using cold-pressed pumpkin (PSF) and okra (OSF) seed oil waste powder, which is also rich in fiber and protein, as flour. Wheat noodles are the second most consumed food after bread worldwide [5] because of their low cost, availability of a variety of products based on texture and taste, long shelf life, and ready-to-eat accessibility. Noodles usually consist of wheat flour, egg, salt, and water. Therefore, based on the consumer demand for a healthy diet, the lack of dietary fiber, vitamin, and protein content in gluten-free noodles needs to be considered, and its formulation may be fortified with dietary fiber-, vitamin-, and protein-rich ingredients [6]. The process of noodle production involves mixing ingredients, sheeting the dough, pre-drying, cutting, and drying [7]. The final quality of the product (approximate composition, color, and polyphenol oxidase activity) depends on the ingredients used. There is only one study in which fruit seeds (grape, pomegranate, and rosehip) were used as functional ingredients in noodle formulation [5], and several studies used different functional ingredients to enrich its protein content such as oats, quinoa, lentils, and edible insects [8,9,10].

The demand for gluten-free products is increasing owing to the increased prevalence of coeliac disease and/or gluten sensitivity [11]. Gluten is an important protein that creates noncovalent interactions and disulfide bonds between gliadins and glutenins to provide the elastic network of doughs [12]. Therefore, it is an essential compound for the desired structure (e.g., less stickiness, unique viscoelasticity of dough, and greater firmness) of bakery products such as bread, pasta, and noodles. The substitution of wheat-based products with gluten-free flour is not able to generate a pseudo-wheat gluten network, especially in terms of sensory properties, appearance, and texture. To obtain the desired features of wheat-based food products, recent studies suggested additional ingredients such as hydrocolloids, proteins, and/or emulsifiers and different dough treatments (enzymatic and physical) [12,13]. The valorization of ESOE waste and the recycling of its byproducts, PSF and OSF, as noodle ingredients may raise the noodles’ dietary fiber and protein content and may also provide the desired pseudo-wheat gluten network with expected textural, physical, and sensorial properties. Pumpkin with seeds is part of the Cucurbitaceae family [14]. The seed of the pumpkin is considered an agro-industrial waste. It is usually cold-pressed to produce phytosterol-, squalene-, phospholipid-, vitamin-, and provitamin-rich seed oil [15]. This process produces a byproduct (seed cake-PSF), which is used as a functional ingredient in bakery products because of its high crude protein (35–53%) and dietary fiber (12–21 g/100 g) content [15,16,17,18]. Okra (*Abelmoschus esculentus*) is one of the most heat- and drought-tolerant flowering plant members belonging to the Malvaceae family [19]. Okra seeds comprise approximately 13.5% of dried okra and are considered agro-industrial waste (OSF) [19]. These seeds are rich sources of dietary fibers (31–41%) and proteins (23–55%) [19,20,21].

However, when compared to products based on wheat flour, most gluten-free products that are available on the market tend to exhibit subpar cooking and sensory qualities [22,23]. To achieve high-quality gluten-free products using alternative ingredients, the use of well-balanced formulations and appropriate technological processes are vital to address the changes in textural and sensorial properties that arise due to the absence of gluten [24,25,26]. Therefore, within the current study, optimization of the noodle formulation was performed by considering the sensory analysis scores of noodles. Hence, the amount of each ingredient needed to be adjusted to achieve the most optimal combination of multiple sensory attributes. To achieve this, a genetic algorithm (using the elitist non-dominated sorting genetic algorithm proposed by Kalyanmoy [27], which is a multi-objective optimization method) was employed while considering numerous input variables (amount of ingredients in noodle formulation), constraints, and objectives (sensory properties).

To sum up, seed cake wastes (PSF and OSF) are an important byproduct of the cold oil press industry to valorize in line with the environmentally friendly use with recycling of waste and the circular economy approach. Both PSF and OSF may provide high added value as active ingredients. The objective of this study was to evaluate the nutritional composition of PSF and OSF as additions to fresh handmade noodles. The noodle formulation was developed employing a genetic algorithm in the R programming language to achieve the most optimal sensory attributes. In addition, physical (cooking properties, color), textural, and sensorial profiles were assessed.

## 2. Materials and Methods

### 2.1. Materials

Cold-pressed oil byproducts of pumpkin and okra cakes were provided from Glokim Bitkisel Yaglar (Melikgazi, Kayseri, Turkey) as flour. Whole fresh eggs, salt, and gluten-free flour mix were purchased from the local supermarket in Isparta, Turkey. The content of the gluten-free flour (Schär (Dr. Schar Turkiye Gida Ltd., İstanbul, Turkey)) consisted of maize starch, rice flour, maize flour, guar gum, and dextrose.

### 2.2. Physicochemical Analysis for Pumpkin and Okra Seed Powders and Noodles

The standard gravimetric method was used for moisture determination. The samples were placed in a drying container that was equilibrated at 105 °C in a drying oven (FN-500, Nuve, Ankara, Turkey) overnight. The moisture content of samples was analyzed according to AOAC, 2005.

Total fat, total protein, salt, and ash values were determined according to the Association of Official Analytical Chemists (AOAC, 2000) for pumpkin and okra seed powders and noodles.

Dietary fiber extraction in pumpkin and okra seed powder and enriched noodle samples were produced based on the method developed by Tejaeda-Ortigoza and colleagues with slight modifications [28]. Before extraction, the samples were treated with hexane to purify the oily and waxy substances that may be present in the milled samples. Subsequently, the water-soluble dietary fibers in the samples were be extracted by stirring them at a rotational speed of 600 rpm in water at 80 °C for 6 h. Although the filtrate contained water-soluble dietary fibers, the pellet contained water-insoluble fibers. For water-insoluble fibers, 50 mM ammonium oxalate and 5 M potassium hydroxide solutions were used. The pH values of the filtrates obtained as a result of the filtrations were adjusted to 7 using 6 M NaOH/HCl solutions, and cold ethanol (stored at −20 °C) was added in an amount 3 times its volume and kept at +4 °C for 12 h. Then, the part containing the precipitated dietary fibers and the aqueous part were separated by using the filtration method (Whatman No: 41). Solid particles were dried at 60 °C for 24 h with the help of a vacuum oven, and dietary fibers were obtained.

### 2.3. Development and Optimization of Noodle Formulation

The recipe of the Süfer was followed for gluten-free noodle production with a slight modification [29]. PSF, OSF, and GFF in amounts of 100 g were mixed at different rates with 40 g of whole egg, 0.5 g of table salt, and finally, 20–23 mL of water, which was added based on dough consistency. To determine the optimum formulation of the gluten-free noodle, 100% PFS, 100% OSF, 100% GF, F, and flour mix were also substituted at different rates between 0–100% based on the model response except for egg content (Table 1). All ingredients were mixed and kneaded with a hand-cranked pasta machine at 2 mm thickness. Then, this dough was cut into pieces 3.5 cm in height and left to dry at room temperature (25 °C). The drying process was stopped at 24 h for all samples and stored at 4 °C in vacuumed polyethylene bags.

#### Sensory Analysis

Gluten-free noodles prepared with different rates of flours, salt, and water were given a sensory evaluation for overall acceptability, hardness, stickiness, taste, chewiness, and color on a 5-point hedonic scale from 1 = extremely bad to 5 = most excellent by 21 trained panelists. The panelists consisted of academicians, postgraduate students, and administrative staff of Suleyman Demirel University, who took a brief lesson about drinking water between tastings, swallowing, chewing, and the time required for this process [29,30]. Noodles were served on a white plate that was coded with three random, single-digit numbers. Between samples, panelists’ mouths were rinsed with distilled water.

Noodle formulation optimization is a multi-objective optimization (MOO) problem. It has several objective functions, all of which share the same decision variables and engage in interaction. The outcome of a single objective optimization or a single objective optimization combined with multi-objective optimization is difficult to judge objectively. As a result, the MOO of the tested noodles was carried out by maximizing all sensory responses using the enhanced nondominated sorting genetic algorithm (NSGA-II) [27]. Moreover, the NSGA-II is fast to convergence and known for its high performance for MOO.

The fundamental idea behind the NSGA-II is that the algorithm starts with a population of decision variables that are randomly chosen. The parent set for the following generation is the decision variable value that produced a better answer. The cycle is repeated until a change has a detrimental impact on one of the objectives [31]. However, the steps taken to perform MOO can be summarized as (i) defining the problem, (ii) executing the NSGA-II, and (iii) analyzing the outcomes. For all data analysis procedures, including the implementation of the MOO, the study utilized the R programming language (release 1.4.1106) and the RStudio (version 1.4.1106) integrated development environment. The NSGA-II was implemented using the ***nsga2()*** function in ***nsga2R*** package [27]. To assess the effectiveness of the default setting, each parameter was altered in an attempt to enhance the Pareto solution set, and population size, number of generations, crossover probability, crossover distribution index, mutation probability, and mutation distribution index were set to 500, 100, 0.7, 5, 0.2, and 10, respectively. The pursuit of a singular objective in real-life problems often conflicts with other objectives, leading to suboptimal outcomes. Hence, achieving a multi-objective solution that simultaneously optimizes each objective is challenging. Instead, a viable approach is to investigate a set of solutions that meet the objectives at an acceptable level without being dominated by any other solution [31]. Therefore, to identify the optimal formulation, a visual inspection of 3D graphics created from the Pareto frontier was performed, with a focus on selecting the combination that was closest to the Pareto optimal point, giving more attention to the variables which were less likely to reach maximum sensorial scores levels.

To use the NSGA-II, it is necessary to mathematically define the objectives based on the decision variables, meaning that altering the decision variable values affects the objectives. The multivariate regression method was used to develop the functions that describe how the objective variables change. To do it, the experimental data were fitted to a full quadratic regression equation (Equation (1))
(1)$$z=\beta_{0}+\overset{n}{\underset{i=1}{\sum }}\beta_{i}x_{i}+\overset{n}{\underset{i=1}{\sum }}\beta_{ii}x_{i}^{2}+\overset{n-1}{\underset{i=1}{\sum }}\overset{n}{\underset{j=i+1}{\sum }}\beta_{ij}x_{i}x_{j}$$
where z is the sensorial response variables of interest; β_0_ is the intercept; β_i_, β_ii,_ and β_ij_ are the coefficients of main effect, quadratic, and interaction terms, respectively; x_i_ and x_j_ describe the independent explanatory variables that correspond to the amount (g) of ingredients in noodle formulations. Multiple, adjusted, and predicted coefficients of determination (R^2^, R^2^_adj_, and R^2^_pred_, respectively) were used for final model evaluation and adequacy. The experimental data were fitted to Equation (1) using the ***lm()*** function in R. The models were simplified by removing non-contributing parameters using mixed stepwise elimination according to the Akaikes’ Information Criterion (AIC) using the ***step()*** function [32]. The lack of fit values for each model was calculated using the ***alr3*** package [33]. The SRC was calculated using the ***std.coef()*** function found in the ***MuMIn*** package [34]. The final regression models, which were used for MOO, are listed in Table 1, and the models’ and regression coefficients’ significance, as well as goodness of fit parameters, are available in the Appendix A. Moreover, standardized model coefficients (SRCs, also known as β-coefficients) are presented to objectively compare the relative effects of the explanatory variables on the outputs.

After developing regression models that describe how noodle formulation affects sensory responses, they were applied to building the optimization problem. Given that high sensory analysis scores indicate high consumer appreciation, it was determined that the objective was to maximize all sensory analysis scores simultaneously. The optimization algorithm was implemented with constraint conditions, which are listed below:(2)X=x∈ℚ+: ∑xi≤50g, ∀i∈xOSF,xPSF,xgff∧0.25g≤xs≤0.5g∧10g≤xw≤13g
(3)Z=z∈ℚ+: 0≤z≤5

### 2.4. Quality and Cooking Parameters of Noodles

#### 2.4.1. Determination of Optimum Cooking Time, Swelling Power, Water Holding Capacity, and Cooking Lost

The method of AACC 66-50.01 (2010) was used to detect the cooking quality and time for the noodles [35]. After the detection of optimum cooking time, the cooked noodles were filtered to remove water and weighed. While the cooked noodles dried at 40 °C for 12 h, the remaining water was dried at 105 °C in a beaker. The cooking loss, water holding capacity, and swelling index were calculated based on Equations (4), (5), and (6), respectively. From the equations, m_1_ represents the weight of the cooked noodles, m_2_ is the weight of the uncooked noodles, and m_3_ indicates the weight of the cooked and dried noodles.
(4)$$\mathrm{cooking}\;\mathrm{loss}\;\left(\%\right)=\frac{m2}{m1}X100$$
(5)$$\mathrm{water}\;\mathrm{holding}\;\mathrm{capacity}=\left(\frac{m2-m1}{m1}\right)X100$$
(6)$$\mathrm{swelling}\;\mathrm{index}=\frac{m2-m3}{m3}$$

#### 2.4.2. Color

Color parameters (L*, a*, and b*) of the noodle samples were measured by using a colorimeter (NH310 High-Quality Portable Colorimeter, Shenzhen 3NH Technology Co., Ltd., Shenzhen, China). According to these values, L* refers to brightness (0: black, 100: white), +a* values represent redness, a* values indicate greenness, +b* values display yellowness, and b* values show blueness. The color values were detected according to the average of at least 10 measurements. Furthermore, the outcomes were employed to determine the complete color variation (∆E) utilizing the prescribed equation (Equation (7)).
(7)$$\Delta E=\sqrt{\Delta L*2+\Delta a*2+\Delta b*2}$$

#### 2.4.3. Total Phenolic Content (TPC) and Antioxidant Activity

The method Toor and colleagues [36] used was applied to measure TPC and antioxidant activity, and ABTS was detected according to the method used by Re and colleagues [37]. The total amount of phenolic substance in the extract was expressed as mg gallic acid equivalent (GAE) per 1 g of dry matter, and the results of the ABTS assay were calculated as %inhibition.

#### 2.4.4. Texture Analysis

The textural properties of the cooked noodle samples were investigated by using the texture profile analysis method of Larrosa et al. [38]. Analysis was performed using a TA-XT Texture Analyzer (TA-XT Plus, Texture Stable Micro Systems, Godalming, UK). Gluten-free noodles were prepared and cooked according to optimal conditions. Cooked noodles were cooled in running water and drained. Texture profile analysis was performed along two compression cycles. A flat-ended aluminum cylindrical probe with a 50 mm diameter (P/50) was used. Five noodle strips were placed on the surface of the analyzer. The test speed was 0.5 mm/s, and the strain was set to 30%. Hardness, adhesiveness, springiness, cohesiveness, and resilience values were obtained from the time–force curve. At least five measurements were taken, and the results are expressed as mean ± standard error.

#### 2.4.5. Scanning Electron Microscopy

A scanning electron microscope (FEI; Quanta 250 FEG, Czech Republic) captured images of raw noodle specimens with suitable (×1000, ×3000, ×5000 or ×10,000) magnifications. After applying a 3 nm-thick coating of Au-Pd, SEM micrographs were performed using 10.0 kV accelerating voltage.

### 2.5. Statistics

All data are presented as means ± SD (standard deviation). Using the statistical program SPSS 20.0, the obtained data were statistically evaluated.

## 3. Results and Discussion

The results of this research show that the dough structure of the gluten-free noodle can be formed as desired, like a gluten dough structure, by using protein and fiber-rich flour. Different studies also reported that additional ingredients such as hydrocolloids, proteins, and/or emulsifiers in gluten-free products may be able to generate a pseudo-wheat gluten network, especially in terms of sensory properties, appearance, and texture [9,12]. For this reason, this study’s results will supply useful information related to the utilization of protein and fiber-rich byproducts of cold-pressed seed oil in gluten-free noodle formulation.

### 3.1. Physicochemical Properties and Proximate Composition of Flours

The physicochemical properties of PSF and OSF are shown in Table 2. The moisture content of the PSF was slightly higher than that of the OSF. In addition, the moisture content (%) of OSF was reported as 7.70 and 7.80 in earlier studies [19,21]. In the current study, cold-pressed okra oil cake was used as an ingredient, and therefore, it may have had lower moisture content than reported in the literature. This reduction in the moisture rate was explained by Akinoso and colleagues in that during the cold-pressed oil process, the heat of the extraction process reduces moisture [39]. The moisture content of the PFS agreed with a previously published study that revealed cold-pressed pumpkin oil cake has 6% moisture content [17]. The total protein, fat, and salt content of the PSF were higher than those of the OSF. The protein and fat content of the PSF were similar to the literature, and they were between 35 and 53% for protein and 14 and 23% for fat content [15,16,17,18]. This is the first study about cold-pressed okra seed oil byproducts used in a new functional product. The moisture content of OSF in the literature varies between 2.7 and 13.9%, which resembles the present study [40]. The protein and fat content of the OSF were also compatible with findings in the literature at ranges of 23 to 55% and 14 to 31%, respectively [19,20,21,40]. The ash and total fiber content of OSF were indicated to be between 3.42 and 9% for ash and 30.8 and 41% for total fiber content in the literature [19,21,40]. The fiber and ash content of the current study’s results are also in agreement with the literature. Therefore, the replacement of pumpkin and okra cold oil-extracted seed cakes with gluten-free flour will help increase the protein and dietary fiber content of gluten-free noodles. Furthermore, the antioxidant activity of both flours was also examined. The TPC amounts in raw okra and pumpkin seeds were found to be 522.47 and 3131.20 mg GAE/100 g, respectively [21,41]. Saykova and colleagues reported that during the cold-pressed oil extraction process, approximately 37–48% of the phenolics and flavonoids remained in the byproducts [42]. The results of the current study show that the TPC of the OSF and PSF were 16.5 and 19.1 mg GAE/100 g, respectively. Due to the process conditions (e.g., there were no oxidation reactions and/or high-temperature applications), phenolic compounds were not damaged and remained within the oil part [43]. Therefore, the TPC content of the byproducts was lower than in the raw materials.

### 3.2. Development of Noodle Formulation

The produced noodles consisted of wheat flour, egg, salt, and water. The optimized formulation for noodle production was achieved based on the MOO method by considering the sensory analysis scores of the noodles. The flour, salt, and water were added to the noodle formulation at different ratios, and the optimized formulation was determined for noodle production. The optimized noodle formulation consisted of OSF, PSF, gluten-free flour, and salt in the following amounts: 11.5, 87.0, 0.9, and 0.6 g, respectively, with 10.5 mL of water. The egg amount in the noodle was constant, and for all formulations, 40 g were added to the mixtures.

As the first step of noodle formulation development, multivariate regression models were fitted to construct the optimization problem. To accomplish this, regression analyses were carried out for nine response variables with five explanatory variables using a full quadratic regression equation. The resulting models were then simplified using mixed stepwise elimination to minimize AIC values. The final regression model provided parameter estimates for both the explanatory formulation variables and the response variables (sensory scores), which can be found in Table 1. For the sake of simplicity, not all of the model’s statistics are given in Table 1; however, more detailed statistics including standardized regression coefficients, models, and parameters’ significance, R^2^, R^2^_adj_, and R^2^_pred_ are available in the Appendix A (please see Appendix A). Please note that the lack of fit statistics of the models were not provided in this table, as they were not found to be significant for any of the fitted equations. To compare the impact of explanatory variables in a model, the use of standardized regression coefficients is more appropriate than using regular regression coefficients. This is because standardized coefficients provide values that are independent of the magnitude and range of the input variables. As a result, variables with different magnitudes and ranges can be compared with each other. The direction of the coefficient, whether negative or positive, indicates the direction of the effect on the output variable. The standardized regression coefficients presented in Appendix A fall outside the typical range of −1 to +1. This is something not generally expected, but including variables with differing scales in a single equation can result in large beta coefficients, even in the thousands. Moreover, in the present study, the standardized regression coefficients were only used for comparing the relative impact of explanatory variables. Thus, they did not affect optimization efforts.

Appendix A shows that all of the first-order terms of the explanatory variables (except salt and partially water) had an important impact on the sensory responses. Specifically, the coefficient representing okra seed and pumpkin concentration in noodle formulation showed a positive impact except for color homogeneity and stickiness. This indicates that higher levels of okra seed and pumpkin in the formulation helped to improve noodle formulation in general. The effect of water and gluten-free flour, on the other hand, was not as strong as those of okra seed and pumpkin. Higher water content also improved sensorial scores except for color homogeneity, stickiness, and overall acceptability. Regarding the use of gluten-free flour, it is worth noting that most gluten-free products available on the market typically demonstrate inferior sensory quality and cooking properties [22,23], and in our study, it was also found to have a negative impact on sensorial analysis scores in general. Furthermore, the second-order terms of okra seed, pumpkin, water, and gluten-free flour also had a significant impact on the predicted values in the developed regression models. For all of the response variables of interest, the second-order terms had the opposite sign of the corresponding first-order terms, indicating a balancing effect on model predictions. However, upon closer inspection, it was observed that the magnitude of the second-order terms did not exceed the real effect of the first-order terms. Thus, when interpreting the effect of model parameters on outputs, evaluations can be made on first-order terms to determine the direction of their impact.

After obtaining the regression equations, a MOO was carried out using the NSGA-II to maximize all sensory responses (for details, please see Section 2.3. Development and Optimization of Noodle Formulation). Using the NSGA-II, three matrices were generated when solving the optimization problem: the non-dominated objectives space, the parameters space, and the Pareto optimal space (a matrix of Booleans indicating whether the solution of each generation in the population is Pareto-optimal (non-dominated)). Figure 1 presents the non-dominated objective space of solutions generated by the NSGA-II to maximize the sensory attributes of the noodle formulation. The figure highlights a strong correlation between some of the sensory attributes, which suggests that maximizing one attribute can lead to the improvement of other sensory properties simultaneously. For instance, increasing the cooking homogeneity score can also improve the chewiness score, and better color homogeneity can lead to improved stickiness in the noodle samples. However, there are instances where this correlation is not so apparent or where there is even a negative correlation between some of the sensory attributes. For example, improving chewiness scores may require compromising the color homogeneity and stickiness of the noodle product. Therefore, obtaining the highest possible values for all sensory scores simultaneously may not be possible. At this point, posterior multi-criteria decision-making becomes crucial to identify the optimal formulation by prioritizing some of the selected sensory properties and compromising on others.

There are several alternatives for selecting the optimal points among the non-dominated solutions produced by evolutionary algorithms, including numeric ones. However, it is also possible for the decision maker to make a trade-off between the set of optimal solutions, which are equally optimal, using high-level information and pick one of the non-dominated solutions as the optimal point depending on the specific problem and its goals, priorities, and constraints [31,44]. That is why the optimal point was determined via visual inspection of the objective space and considering the statistics given in Table 3. According to the data presented in this table, the lowest values that were obtained for aroma and pleasant taste were greater than 4.5 for both. These levels were considered to be already sufficient, and it was decided that there was no need to prioritize these two sensory scores for optimization. The maximum forecasted score of overall acceptability, on the other hand, was 4.38, which is considerably lower than the highest values of the other sensory properties. Because overall acceptability provides an approximate score for the general approval of the product after the evaluation of all sensory criteria by the panelists, it was considered to be a parameter that is required to be prioritized for maximization. The lowest minimum values in the objective space and the largest variations were found in color homogeneity and stickiness among the remaining parameters. However, it can be observed from Figure 1 that there is a positive correlation between these two variables, indicating that maximizing one of them could also result in high values for the other. Based on the results of verbal interviews with the panelists, it was decided that color homogeneity was a more important parameter, and thus, it was included as one of the variables that should be prioritized while maximizing, following overall acceptability. Moreover, there is a positive correlation among the remaining sensory parameters (color, cooking homogeneity, chewiness, and unpleasant taste), which can be seen in Figure 1. This means that maximizing one of these parameters will have a positive effect on the others. Because chewiness had the lowest variance and the highest minimum value among these parameters, its maximization was considered to be easier. Thus, it was prioritized as the third response that needs to be optimized.

When the non-dominated solutions for these three prioritized parameters were plotted in a 3D space, Figure 2 was obtained. The combination where all three response variables reach their maximum value was selected on this plot and is considered to be the optimal point (the selected point is marked in Figure 2). At this point, the estimated optimum levels of the other sensory parameters are given as fitted values in Table 3. As can be seen, except for unpleasant taste, all other parameters are expected to have sensory quality levels above 4. The prediction and confidence intervals (at a 95% level for both) for these variables were also presented in the same table. (Please note that the maximum score in the real-life panel was 5. Therefore, any model predictions that were greater than 5 were rounded down to 5.) The ingredients in the formulation necessary to achieve the predicted sensory properties at the optimal point were calculated to be 7.42% okra seed, 54.06% pumpkin, 0.40% salt, 12.64% water, 0.55% gluten-free flour, and 24.92% egg on a mass basis.

The optimized formulation was then used to produce new batches of noodles, and three independent sensory analysis panels were conducted as described in the Materials and Methods section. The scores obtained from the panelists in different panels are presented in Table 3. Although there were some negligible differences among the results obtained from different panels (and it is quite expected to find some variations between the scores obtained from different panels), all of the sensory analysis results from the validation tests were within the predicted ranges and/or confidence intervals. This demonstrates the accuracy and repeatability of the results obtained from this formulation optimization study.

### 3.3. Physicochemical Properties and Approximate Composition of Noodles

Consumers pay great attention to the cooking qualities of the noodle. The cooking parameters of the noodle samples in terms of optimum cooking time, swelling index, water-holding capacity, volume increase, and cooking loss are shown in Table 4. The determination of water-holding capacity is important for gluten-free products, as it is a parameter of the viscoelasticity of the batter and dough, its gelatinization and pasting behavior, and the final product’s texture properties [10,45]. In addition, the carbohydrate fractions and protein content as well as their possible interactions influence the water-holding capacity of the flour [10]. The optimized formulated noodle has a water-holding capacity of 107%, and its swelling power was 1.87 g/g. The swelling power of other gluten-free noodle production studies ranged between 9 and 19 g/g [10]. The current product’s swelling power was lower than reported in the literature, which may be related to its protein content. Joshi and colleagues reported that the protein content of the flour may prevent the swelling of starch because proteins may absorb the water in the product [46]. The optimum cooking time for the noodles was found to be 35 min (Table 4). In the literature, there are different cooking time values for noodle products, reported with values between 4.5 min and 23 min [5,10,47,48,49,50]. However, the production of these previously studied noodles involved some pretreatments (e.g., steam-jet cooking, roasting, or extrusion), and their thickness and diameter values are smaller than the noodles of the current study. Specifically, a rise in protein composition results in a reduction of adhesiveness in spaghetti, whereas higher protein content leads to less variation in the physical attributes of spaghetti during the cooking process, resulting in increased resistance to overcooking [51]. Yao and colleagues (2020) also reported that noodles with high protein content cooked slower than those with low protein content. In addition, it was also reported that noodles that possess elevated levels of protein exhibit a considerable increase in textural attributes such as hardness, springiness, chewiness, and adhesiveness compared to other rehydrated noodles, particularly in the cases of chewiness and adhesiveness. Therefore, in the current study, the cooking time may have been longer than as reported in the literature based on the high protein content of the noodles [52]. The strength of the protein network and cooking loss is expected to be parallel. If this network is not sufficient, the nutritional quality of the product will be reduced due to the leak of water-soluble solids and minerals during cooking [23,49]. The acceptable cooking loss limits for durum wheat pasta are between 7 and 8% [10,53]. In addition, in other studies, it was indicated that gluten-free noodles or kinds of pasta prepared with oat, buckwheat, quinoa, amaranth, yellow lentil, and chickpea have cooking loss rates between 4.7 and 13% [9,49,53]. The noodles produced in the current study had a cooking loss value of 6.7%, and it is within the limits of the literature.

Recent studies have paid attention to deficient fiber and protein content for gluten-free noodles or pasta, as they are mostly composed of starch-based flour [49,50]. The proximate composition of these noodles was detected as follows: TP (42.88%), TF (15.6%), ash (5.68%), TDF (40.48%), TPC (25.5 mg GAE/100 g), and ABTS (406.09 ± 12.78 mg TE/100 g). In addition, the results demonstrate that the utilization of cold-pressed pumpkin and okra seed flours for gluten-free noodle formulations considerably improve the protein and fiber content better than durum wheat pasta [53]. Therefore, valorization of these byproducts as an ingredient may increase the interest of both consumers and processors and may support their valorization in line with the goal of environmentally friendly use via recycling waste and the circular economy approach.

The color of the noodle was used as another factor to determine consumers’ preference, and the results are given in Table 5. Different studies indicated that the color of fortified products plays an important role in the final product’s color [9,19,21]. The results show that the noodle has a yellowish color (b*) with moderate lightness (L*). The color difference between cooked and uncooked noodles (ΔE) was determined to be 5.75 ± 0.53. Moreover, the rate of PFS in the optimized noodle was higher than OSF; therefore, the final product had slightly higher greenness values (a*) than OSF.

The surface morphology of the noodles was determined via SEM to observe the network structures in the noodles (Figure 3). According to the SEM images of the noodles, they had inhomogeneous porous structures, and proteins may have prevented the complete gelatinization of the starch, as the noodles exhibited compact and dense network morphology in terms of less cooking loss and higher cooking time.

The effect of the incorporation of PSF and OSF into noodle samples on TPA is presented in Table 6. The hardness of noodles was directly affected by ingredients in the formula (e.g., eggs, proteins, starches, and hydrocolloids, etc.), which demonstrates an influence on the cooked noodle [10,54,55]. Protein quality and amount showed an effect on noodle hardness based on the protein matrix’s strength and integrity [56].

The cooked noodles’ integrity is another important consumer desire, and without a gluten network, it may be hard to protect it. Adding protein-rich ingredients to gluten-free noodles results in the consumers’ desired dough firmness [50]. The adhesiveness is related to the release of starch from the noodles during cooking, and it directly affects the amount of loss during cooking [10]. The chewiness value is also affected similarly to the hardness value due to its protein network. The chewiness value shows the required energy to disintegrate the noodles before swallowing, and higher chewiness values for cooked noodles may be considered for better homogeneity and a less crumbly texture [55].

## 4. Conclusions

In this study, the valorization of cold-pressed pumpkin and okra seed byproducts (cake) for noodle production was investigated for the first time. In addition, the results of the study indicate a practical and effective method to develop a protein- and fiber-enriched gluten-free noodle formulation by utilizing a statistical mixture design of experiments. The development of an optimized noodle formulation using a multi-objective optimization approach is presented using a multi-objective optimization approach (the enhanced nondominated sorting genetic algorithm (NSGA-II)) to maximize all sensory responses. The optimized noodle formulation consisted of byproducts of cold-pressed pumpkin and okra seeds, gluten-free flour, salt, and egg with specific amounts (54.06, 7.42, 0.55, 0.40, and 24.92% on a mass basis, respectively). A set of multivariate regression models were used to fit the optimization problem and provide parameter estimates for both the explanatory formulation variables and the response variables (sensory scores). The results showed that the concentrations of okra and pumpkin seeds in the noodle formulation had a positive impact on sensory attributes. In contrast, the use of gluten-free flour had a negative impact. Both seed cakes are rich in terms of dietary fibers and proteins. These results provide an opportunity to add sustainable monetary value to cold-pressed oil cakes by converting pumpkin and okra seed cakes into food ingredients that add high value. The nutritional potential of the developed noodles has approximately 43% protein and 40% dietary fiber content. Moreover, according to textural, physical, and sensorial properties, the final product is able to demonstrate the expected pseudo-wheat gluten network. Finally, these results suggest that pumpkin and okra seed cakes could improve the nutritional and sensory quality of gluten-free noodles in terms of protein and dietary fiber.

## Figures and Tables

**Figure 1 foods-12-02018-f001:**
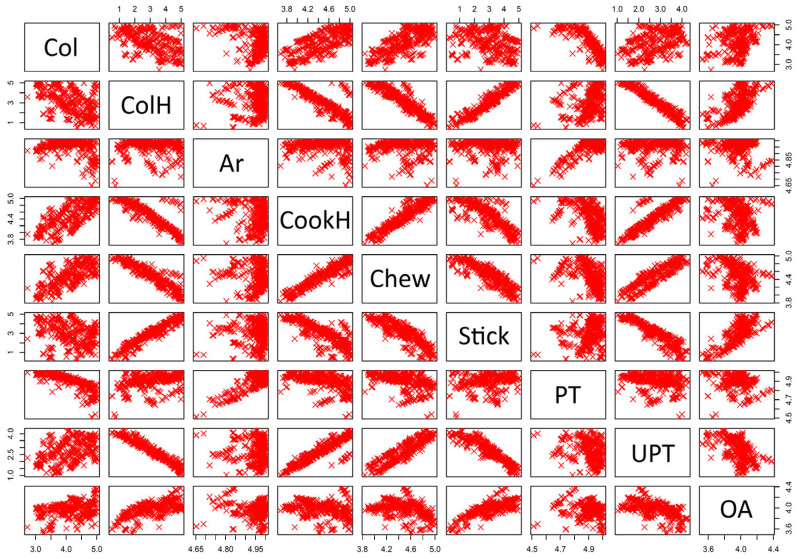
Non-dominated objective space of the solutions produced by the NSGA-II for maximizing the sensory attributes of the noodle formulation (Col: color, ColH: color homogeneity, Ar: aroma, CookH: cooking homogeneity, Chew: chewiness, Stick: stickiness, PT: pleasant taste, UPT: unpleasant taste, OA: overall acceptability).

**Figure 2 foods-12-02018-f002:**
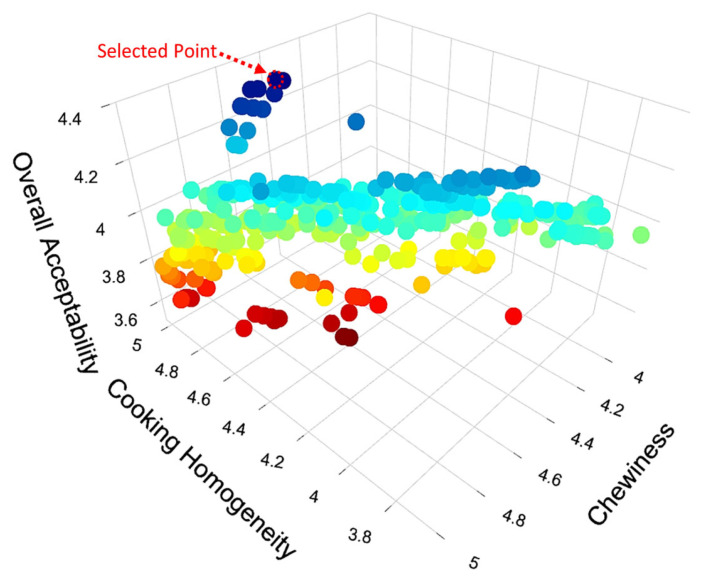
Non-dominated solution for overall acceptability, cooking homogeneity, and chewiness (The colors represent the range of overall acceptability for non-dominated solutions, with lower values shown in red to higher values shown in blue).

**Figure 3 foods-12-02018-f003:**
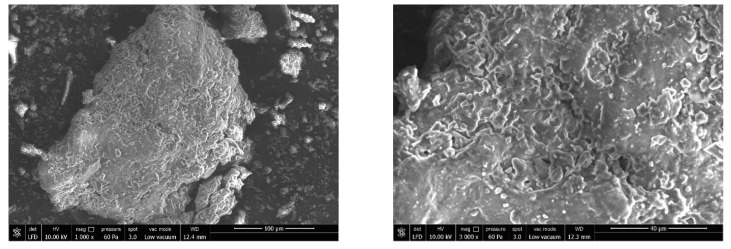
Scanning electron micrographs of noodles at 1000, 3000, 5000, and 10,000× magnification.

**Table 1 foods-12-02018-t001:** Multi-objective optimization model coefficient terms added after mixed stepwise elimination in accordance with the Akaikes’ Information Criterion.

	*z_Col_*	*z_ColH_*	*z_Ar_*	*z_CookH_.*	*z_Chew_*	*z_Stick_*	*z_PT_.*	*z_UPT_*	*z_OA_*
** *β* _0_ **	−3529.37	4478.85	−674.27	−2446.59	−2139.56	3180.06	−415.33	−4613.85	−769.94
** *β_OSF_* **	204.09	−248.84	39.67	139.93	121.95	−167.89	24.67	261.73	46.90
** *β_PSF_* **	204.28	−249.06	39.74	140.04	122.05	−167.94	24.74	262.06	46.97
** *β_s_* **	24.64		8.40	11.96	12.67	8.93	6.91	17.21	11.30
** *β_w_* **	17.84	−55.93	1.24	18.13	16.75	−68.01		39.81	−3.73
** *β_gff_* **	−92.53	113.07	−17.84	−63.48	−55.27	76.57	−11.05	−118.62	−21.14
**(*β_OSF_*)^2^**	−2.72	3.32	−0.53	−1.86	−1.63	2.24	−0.33	−3.49	−0.62
**(*β_PSF_*)^2^**	−2.72	3.33	−0.53	−1.87	−1.63	2.24	−0.33	−3.50	−0.63
**(*β_w_*)^2^**	−0.62	2.33		−0.70	−0.63	2.95	0.04	−1.56	0.23
**(*β_gff_*)^2^**	3.21	−3.92	0.62	2.20	1.92	−2.65	0.38	4.12	0.74
** *β_OSF_β_PSF_* **	−5.44	6.64	−1.06	−3.73	−3.25	4.48	−0.66	−6.98	−1.25

*β_i_* terms represent the multivariate regression models’ coefficients (given in Equation (1)) that were used to define the relation between objective (zi) and explanatory variables. The subscripts of *β_i_* terms are the following: 0: intercept, OSF: okra seed, PSF: cold-pressed pumpkin seed, s: salt, w: water, and gff: gluten-free flour. The subscripts of *z_i_* terms are the following: Col: color, ColH: color homogeneity, Ar: aroma, CookH: cooking homogeneity, Chew: chewiness, Stick: stickiness, PT: pleasant taste, UPT: unpleasant taste, and OA: overall acceptability.

**Table 2 foods-12-02018-t002:** Physicochemical properties and approximate composition of okra (OSF) and pumpkin (PSF) seed flours.

	Moisture	TP	TF	TC	TDF	Salt	Ash	TPC	ABTS
PSF	5.36 ± 0.12	39.35 ± 0.31	16.68 ± 0.20	7.55 ± 0.17	17.97 ± 0.02	n.a	3.55 ± 0.39	19.1 ± 4.00	296.18 ± 15.16
OSF	5.20 ± 0.34	32.68 ± 0.14	8.46 ± 2.05	21.3 ± 0.24	32.39 ± 0.17	n.a	4.90 ± 0.03	16.5 ± 3.00	218.20 ± 15.51

TP: Total Protein (%), TF: Total Fat (%), TC: Total Carbohydrate (%), TDF: Total Dietary Fiber (%), TPC: Total Phenolic Content (mg GAE/100 g), ABTS: 2,2’-azino-bis (3-ethylbenzothiazoline-6-sulfonic acid radicals (mg TE/100 g).

**Table 3 foods-12-02018-t003:** Statistics of non-dominated objective space of the solutions produced by using NSGA-II, fitted values of the optimized objective variable, and results of validation experiments.

	Statistics of Objective Space			Optimization Results			Validation Scores		
	Min	Max	Var	Fit. Val.	Pred. Int.	Conf. Int.	Trial 1	Trial 2	Trial 3
Col	2.73	4.99	0.32	4.95	4.70–5.20	4.44–5.47	4.64 ± 0.11	4.76 ± 0.22	4.70 ± 0.14
ColH	0.51	4.99	1.63	4.92	4.82–5.02	4.66–5.18	4.79 ± 0.20	4.81 ± 0.15	4.77 ± 0.13
Ar	4.65	4.99	0.01	4.84	4.68–5.01	4.49–5.20	4.71 ± 0.13	4.69 ± 0.16	4.73 ± 0.14
CookH	3.68	4.99	0.14	4.85	4.66–5.03	4.47–5.22	4.79 ± 0.09	4.74 ± 0.11	4.70 ± 0.22
Chew	3.84	4.99	0.09	4.49	4.28–4.69	4.07–4.91	4.53 ± 0.15	4.57 ± 0.22	4.54 ± 0.14
Stick	0.29	4.99	1.57	4.13	3.91–4.35	3.67–4.59	4.04 ± 0.13	4.11 ± 0.27	4.09 ± 0.20
PT	4.51	4.99	0.01	4.81	4.61–5.01	4.37–5.24	4.89 ± 0.12	4.86 ± 0.11	4.87 ± 0.11
UPT	1.03	4.25	0.69	2.02	1.64–2.41	1.22–2.83	2.13 ± 0.11	2.16 ± 0.21	2.14 ± 0.18
OA	3.53	4.38	0.02	4.38	4.27–4.48	4.17–4.59	4.56 ± 0.26	4.49 ± 0.26	4.46 ± 0.31

Col: color, ColH: color homogeneity, Ar: aroma, CookH: cooking homogeneity, Chew: chewiness, Stick: stickiness, PT: pleasant taste, UPT: unpleasant taste, OA: overall acceptability, Var: variance, Fit. Val.: fitted value, Pred. Int: prediction interval, Conf. Int.: confidence interval. Confidence and prediction interval values (level of 95% for both) are given as lower–upper limits. Validation scores are given as mean ± standard deviation. Please note that the maximum score in the real-life panel was 5. Therefore, any model predictions that were greater than 5 were rounded down to 5.

**Table 4 foods-12-02018-t004:** Cooking properties of optimized noodles.

Properties	Optimized Noodles
Optimum cooking time (min)	35.00
Swelling power (g/g)	1.87 ± 0.02
Cooking lost (%)	6.68 ± 0.04
Water-holding capacity (%)	107.60 ± 1.35

**Table 5 foods-12-02018-t005:** Color properties of noodles, PSF, and OSF.

Properties	Uncooked Noodles	Cooked Noodles	PSF	OSF
**L***	63.64 ± 0.33	58.63 ± 0.91	70.64 ± 2.60	63.28 ± 0.40
**a***	0.54 ± 0.02	0.71 ± 0.01	−2.49 ± 0.04	2.41 ± 0.06
**b***	17.61 ± 0.22	19.32 ± 0.46	20.86 ± 0.11	13.56 ± 0.14

**Table 6 foods-12-02018-t006:** Texture analysis of noodles.

	Hardness (g)	Adhesiveness (g.sec)	Springiness	Cohesiveness	Gumminess	Chewiness	Resilience
Noodle	2534 ± 15.8	−0.058 ± 0.01	2621.66 ± 12.5	0.880 ± 0.02	1624.038 ± 18.7	4261.5 ± 33.0	0.646 ± 0.03

## Data Availability

The data presented in this study are available on request from the corresponding author.

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
