# Peer review of "A New Approach for the Development and Optimization of Gluten-Free Noodles Using Flours from Byproducts of Cold-Pressed Okra and Pumpkin Seeds"

_foods, 2023, doi:10.3390/foods12102018_

Round 1
Reviewer 1 Report
Article „Valorization of cold-pressed seed by-products: Development 2
and Optimization of gluten-free noodles ” is interesting because it optimizes the production of pasta in terms of the use of zero waste technology. It is also very important that the authors undertook the study of a gluten-free product.
Unfortunately, there are a few mistakes;
When determining the color of the product, please calculate delta E. i.e (ΔE) - a parameter that numerically defines the difference between the two compared colors, most often the deviation of the color obtained in production from the standard.
Please convert the antioxidant activity marked ABTS to Trolox (TEAC), because the percentage of inhibition is not reliable.
When discussing the culinary characteristics of pasta, the authors should pay attention to how the introduced additive disrupts the primary matrix of gluten-free pasta (control). They should explain how the ingredients (protein, fat, fiber, phenols) derived from addditions (cold press pumpkin (PSF) 14 and okra (OSF) seeds' by-product s) will affect the cooking characteristics and texture of the pasta.
Conclusions too general

Article „Valorization of cold-pressed seed by-products: Development 2
and Optimization of gluten-free noodles ” is interesting because it optimizes the production of pasta in terms of the use of zero waste technology. It is also very important that the authors undertook the study of a gluten-free product.
Unfortunately, there are a few mistakes;
When determining the color of the product, please calculate delta E. i.e (ΔE) - a parameter that numerically defines the difference between the two compared colors, most often the deviation of the color obtained in production from the standard.
Please convert the antioxidant activity marked ABTS to Trolox (TEAC), because the percentage of inhibition is not reliable.
When discussing the culinary characteristics of pasta, the authors should pay attention to how the introduced additive disrupts the primary matrix of gluten-free pasta (control). They should explain how the ingredients (protein, fat, fiber, phenols) derived from addditions (cold press pumpkin (PSF) 14 and okra (OSF) seeds' by-product s) will affect the cooking characteristics and texture of the pasta.
Conclusions too general
Author Response
Dear Professor,
First, thank you very much for your kind and valuable suggestions and corrections. All the corrections were made as per your suggestions, highlighted in the manuscript, and answers are given below:
Best regards
Response to Reviewer 1 Comments
The article „Valorization of cold-pressed seed by-products: Development and Optimization of gluten-free noodles” is interesting because it optimizes the production of pasta in terms of the use of zero-waste technology. It is also very important that the authors undertook the study of a gluten-free product. Unfortunately, there are a few mistakes.
Point 1. When determining the color of the product, please calculate delta E. i.e. (ΔE) - a parameter that numerically defines the difference between the two compared colors, most often the deviation of the color obtained in production from the standard.
Thank you for taking the time to read our study carefully. Please see the lines between 493 and 495 for additional information.
The formula was also added to the methods. Please see the lines between 231 and 233 for additional information.
Point 2. Please convert the antioxidant activity marked ABTS to Trolox (TEAC), because the percentage of inhibition is not reliable.
Thanks for your suggestion, the antioxidant activity was changed to Trolox Equivalent.
Point 3. When discussing the culinary characteristics of pasta, the authors should pay attention to how the introduced additive disrupts the primary matrix of gluten-free pasta (control). They should explain how the ingredients (protein, fat, fiber, phenols) derived from additions (cold press pumpkin (PSF) 14 and okra (OSF) seeds' by-products will affect the cooking characteristics and texture of the pasta.
Please see the additional paragraph given between 461 and 470.
Point 4. Conclusions too general
Please see the additional information for a more detailed conclusion.

Reviewer 2 Report
I revised the manuscript concerning the development of gluten free noodles by using cold press pumpkin and okra seeds' by-product. The results revealed an increase in proteins and dietary fiber in noodle, which also showed good sensory quality. The experiments are performed rigorously and results are sufficiently discussed. In my opinion, the most interesting and innovative aspect of the paper, besides the use of by-products, is the development of an algorithm to predict the best ingredients formulation in terms of sensory attributes nutritional composition, color, cooking, and textural properties. Because this aspect is largely discussed in the paper, I suggest to modify the title and to enrich the conclusions, taking into account this result.
Minor revisions:
Lines 46-47: on the basisi of this sentence we can consider the use of wholegrain flour to preserve fiber, vitamins and others bioactives naturally present in the kernel. Anyway, this research is focused on gluten free noodles hence I suggest modifying it in terms of fiber and protein deficiency in gluten free products.
101-102: specify the ingredients of the gluten free flour
111-112: Please reword this sentence
138: I'm not an expert of algotithm but I ask to the authors if 21 panelists are sufficient to develop the alghorithm in which many variables are present
275: the Authors declare this is the first study on cold pressed okra seed oil by-product but later in the text they report other study on OSF.
Table 2: This paragraph is Physicochemical Properties and Proximate composition of Flours. Why noodles? Their formulation is reported in the follow paragraph
449: 35 min to coock a plate of noodles? It is very energy and time consuming
English should be checked throughout the text
Author Response
Dear Professor,
First, thank you very much for your kind and valuable suggestions and corrections. All the corrections were made as per your suggestions, highlighted in the manuscript, and answers are given below:
Best regards
Response to Reviewer 2 Comments
Suggestions for Authors
I revised the manuscript concerning the development of gluten-free noodles by using cold-press pumpkin and okra seeds' by-products. The results revealed an increase in proteins and dietary fiber in noodles, which also showed good sensory quality. The experiments are performed rigorously, and results are sufficiently discussed. In my opinion, the most interesting and innovative aspect of the paper, besides the use of by-products, is the development of an algorithm to predict the best ingredients formulation in terms of sensory attributes nutritional composition, color, cooking, and textural properties. Because this aspect is largely discussed in the paper, I suggest to modify the title and to enrich the conclusions, taking into account this result.
Minor revisions:
Point 1. Lines 46-47: on the basis of this sentence, we can consider the use of wholegrain flour to preserve fiber, vitamins, and other bioactive naturally present in the kernel. Anyway, this research is focused on gluten-free noodles hence I suggest modifying it in terms of fiber and protein deficiency in gluten-free products.
The corrections were made based on your suggestions.
Point 2. 101-102: specify the ingredients of the gluten-free flour
It was added to the lines between 102 and 104.
Point 3. 111-112: Please reword this sentence
The sentence is corrected, thank you for your attention.
Point 4. 138: I'm not an expert of algorithm but I ask the authors if 21 panelists are sufficient to develop the algorithm in which many variables are present
Thank you for taking the time to read our study carefully. We agree with you that having a larger number of observations is always beneficial for algorithms, especially when dealing with variables such as sensory analysis results that tend to have higher levels of uncertainty. However, the regression models used in our optimization algorithm, as well as the results obtained from validation experiments (in which we compared the optimal recipe with experimental formulations) are both yielded satisfactory results. This suggests that the number of panelists studied was sufficient within the scope of the current study. However, in cases where the level of uncertainty is higher, it is always beneficial to increase the number of panelists.
Point 5. 275: the Authors declare this is the first study on cold-pressed okra seed oil by-product but later in the text they report other studies on OSF.
It was revised as ‘This is the first study about cold pressed okra seed oil by-products used in a new functional product.’
Point 6. Table 2: This paragraph is Physicochemical Properties and Proximate Composition of Flours. Why noodles? Their formulation is reported in the following paragraph
Thank you for your attention, the Physicochemical Properties and Proximate composition of noodles were removed and presented as text between lines 481 and 483.
Point 7. 449: 35 min to cook a plate of noodles? It is very energy and time consuming
It is quite true that 35 min is a long time for cooking a plate of noodles. However, the previously published studies used pretreatments such as steam-jet cooking, roasting, or extrusion. Also, as an alternative study to decrease the temperature, different cooking methods such as ohmic heating may be applied (Turgut et al., 2021). Please also see the lines between 461 and 470.
Turgut, Y., Turgut, S. S., & Karacabey, E. (2021). Use of ohmic heating as an alternative method for cooking pasta. Journal of the Science of Food and Agriculture, 101(13), 5529-5540.
Point 8. English should be checked throughout the text
It is revised based on your suggestion.
